# Acute Pulmonary Embolism in COVID-19: A Potential Connection between Venous Congestion and Thrombus Distribution

**DOI:** 10.3390/biomedicines10061300

**Published:** 2022-06-02

**Authors:** Franck Nevesny, David C. Rotzinger, Alexander W. Sauter, Laura I. Loebelenz, Lena Schmuelling, Hatem Alkadhi, Lukas Ebner, Andreas Christe, Alexandra Platon, Pierre-Alexandre Poletti, Salah D. Qanadli

**Affiliations:** 1Department of Vascular and Interventional Radiology, François-Mitterrand University Hospital, 21079 Dijon, France; brothersfs@hotmail.fr; 2Cardiothoracic and Vascular Division, Department of Diagnostic and Interventional Radiology, Lausanne University Hospital (CHUV) and University of Lausanne (UNIL), 1011 Lausanne, Switzerland; salah.qanadli@chuv.ch; 3Department of Radiology, University Hospital Basel, University of Basel, 4031 Basel, Switzerland; alexander.sauter@usb.ch (A.W.S.); lenahannachristine.schmuelling@usb.ch (L.S.); 4Department of Diagnostic, Interventional and Pediatric Radiology, Inselspital, Bern University Hospital, University of Bern, 3010 Bern, Switzerland; laura.loebelenz@insel.ch (L.I.L.); lukas.ebner@insel.ch (L.E.); 5Institute of Diagnostic and Interventional Radiology, University Hospital Zurich, University of Zurich, 8006 Zurich, Switzerland; hatem.alkadhi@usz.ch; 6Department of Radiology, Division City and County Hospitals, Inselgroup, Bern University Hospital, University of Bern, 3004 Bern, Switzerland; andreas.christe@insel.ch; 7Service of Radiology Division of Clinical Epidemiology Service of Radiology, Geneva University Hospital, 1205 Geneva, Switzerland; alexandra.platon@hcuge.ch (A.P.); pierre-alexandre.poletti@hcuge.ch (P.-A.P.)

**Keywords:** CT, CT-angiography, infection, lung, pulmonary embolism

## Abstract

Background: Vascular abnormalities, including venous congestion (VC) and pulmonary embolism (PE), have been recognized as frequent COVID-19 imaging patterns and proposed as severity markers. However, the underlying pathophysiological mechanisms remain unclear. In this study, we aimed to characterize the relationship between VC, PE distribution, and alveolar opacities (AO). Methods: This multicenter observational registry (clinicaltrials.gov identifier NCT04824313) included 268 patients diagnosed with SARS-CoV-2 infection and subjected to contrast-enhanced CT between March and June 2020. Acute PE was diagnosed in 61 (22.8%) patients, including 17 females (27.9%), at a mean age of 61.7 ± 14.2 years. Demographic, laboratory, and outcome data were retrieved. We analyzed CT images at the segmental level regarding VC (qualitatively and quantitatively [diameter]), AO (semi-quantitatively as absent, <50%, or >50% involvement), clot location, and distribution related to VC and AO. Segments with vs. without PE were compared. Results: Out of 411 emboli, 82 (20%) were lobar or more proximal and 329 (80%) were segmental or subsegmental. Venous diameters were significantly higher in segments with AO (*p* = 0.031), unlike arteries (*p* = 0.138). At the segmental level, 77% of emboli were associated with VC. Overall, PE occurred in 28.2% of segments with AO vs. 21.8% without (*p* = 0.047). In the absence of VC, however, AO did not affect PE rates (*p* = 0.94). Conclusions: Vascular changes predominantly affected veins, and most PEs were located in segments with VC. In the absence of VC, AOs were not associated with the PE rate. VC might result from increased flow supported by the hypothesis of pulmonary arteriovenous anastomosis dysregulation as a relevant contributing factor.

## 1. Introduction

The 2019 coronavirus disease (COVID-19) is a multi-systemic disease caused by the new severe acute respiratory syndrome coronavirus 2 (SARS-CoV-2), for which different clinical profiles have been reported. A patient subset may have acute respiratory distress syndrome (ARDS) with a severe or even lethal disease course [1,2] and/or cardiovascular complications [3,4,5]. From the pathophysiological point of view, interference between pulmonary and cardiovascular consequences, with or without morphologic expression, have been debated [6,7,8].

A better understanding of the factors responsible for hypoxemia may help to improve patient management. In COVID-19-related pneumonia, clinical worsening secondary to hypoxemia appears to correlate with a state of hypercoagulability and pro-thrombotic conditions. The pathogenesis of hypercoagulability observed in COVID-19 is not entirely understood. Multiple factors are supposed to act in parallel, including but not limited to endothelial injury, endotheliitis, microvascular dysfunction, pro-thrombotic factor dysregulation, hyperviscosity, and sepsis. These and additional factors, including critical illness, central venous catheters, immobilization, sedation, and vasopressors, can predispose patients to venous thromboembolic disease (VTE), including both deep vein thrombosis (DVT) and pulmonary embolism (PE) [9]. Autopsy studies found thrombosis in nearly 50% of patients who died following COVID-19, highlighting the pivotal role of PE in severe cases (one third had massive PE and one quarter had DVT) [10]. These findings have led to the intensive and empirical use of thromboprophylaxis in many intensive care units, and despite initial hope, current evidence does not support higher-dose anticoagulation [11].

It has been suggested that a disturbed hematologic environment accompanies severe COVID-19 pneumonia. A potential mechanism proposed is the recruitment of inflammatory cytokine and viral damage to endothelial cells, causing pulmonary microcirculatory thrombosis [12,13] and vascular dilatation [14,15]. Parenchymal involvement in COVID-19 pneumonia has been described extensively; however, data on virus-induced vascular abnormalities (arterial and/or venous) are scarce, especially since most current studies focus on non-contrast CT [16].

The mechanisms governing blood flow changes are challenging to understand. Pulmonary microvascular involvement has been suggested in COVID-19 pneumonia using dual-energy computed tomography (DECT) with pulmonary hyperperfusion in the lobes with alveolar opacity (AO), whereas no perfusion abnormalities were found in normal parenchymal areas [17]. This finding could support the theory that emboli move to areas of increased blood flow. On the other hand, areas of oligemia have been consistently described as well [8,15], which are seemingly contradictory to vascular dilatation suggestive of increased blood flow and blood volume [18]. The simultaneous presence of oligemia and vascular dilatation raises the question of intrapulmonary arteriovenous shunt mechanisms that may play a role in the discrepancy between the morphological appearance on imaging and severe hypoxia [6].

Our primary objective is to better characterize vascular abnormalities in patients with COVID-19 and PE. The prevalence of PE, its severity, and topographical distribution according to the affected lung areas in COVID-19 pneumonia were analyzed, as was the relationship between thrombus location and the presence of vascular dilatation and/or AOs in the same lung segment.

## 2. Materials and Methods

### 2.1. Study Design and Cohort

In this study, we analyzed data from the Swiss COVID-CAVA multicenter registry (clinicaltrials.gov identifier NCT04824313) [19] that included consecutive patients with microbiological evidence of SARS-CoV-2 infection and chest CT, retrospectively collected from five university hospitals and affiliates in Switzerland at the beginning of the pandemic, from March until June 2020. The participating centers were in Lausanne, Zurich, Basel, Bern, and Bern County Hospitals, and all were affiliated with a university.

We screened consecutive COVID-19 patients who underwent clinically indicated CT pulmonary angiography (CTPA) for suspected PE. The inclusion criteria for this study were age > 18 years, a COVID-19 diagnosis confirmed by reverse transcription-polymerase chain reaction for SARS-CoV-2, and a radiological diagnosis of PE through CTPA.

Exclusion criteria were patients with another pre-existing infectious process (*n* = 0), non-optimal CT scan, incomplete CT data (*n* = 10), or documented refusal of the reuse of medical data (*n* = 5). Of the 268 screened patients, 61 (23%) had PE proven upon CTPA and were included for further analysis. Clinical and demographic data were retrieved from electronic records. The sample size estimation was designed for 80% power and 5% type-one error rate. Under the assumption that PE distribution is independent of lung inflammation, we analyzed at least 182 lung segments with PE to detect a 20% increase in PE within segments with AO. Details on the design and rationale were reported in the study protocol [19]. A flow chart of the study population is shown in Figure 1. The protocol was approved by the local Ethics Commission (project-ID 2020-01469).

### 2.2. Data Analysis

#### 2.2.1. Clinical Data

Patient demographics, admission date, comorbidities, clinical course, and laboratory findings were collected from electronic medical record systems. Fatal outcomes and intensive care unit (ICU) admission were registered. We defined composite outcome as either ICU admission or death; this definition of composite outcome has been previously published [3]. The composite outcome was analyzed with respect to airspace disease and pulmonary vascular dilatation.

#### 2.2.2. Imaging Data

A radiologist with five years of experience (F.N.) in thoracic imaging reviewed the CTPA images of the patients included in the study on a PACS workstation (Carestream Health, New York, United States, V12.2.1025). The radiologist was blinded to the patients’ conditions and clinical and biological parameters. The 61 patients with PE were further analyzed segment by segment (yielding a total of 1220 lung segments according to the Boyden classification). Concomitant respiratory or pulmonary diseases were registered, such as malignancy and pleural effusion. Then, we performed qualitative and quantitative image analysis for various parameters detailed hereafter.

#### 2.2.3. Vascular Dilatation

Vascular dilatation was first assessed as a qualitative variable on CT. Specifically, vascular dilatation was defined as previously suggested [15]: “vessel diameter larger than expected for the point within the vascular tree, characterized by: (a) vessel diameter larger than that in adjacent portions of nondiseased lung, (b) vessel diameter larger than that in comparable regions of nondiseased contralateral lung, or (c) focal dilatation or non-tapering of vessels as they course toward the lung periphery”. In a second step to define whether the vascular dilatation phenomenon affected arteries, veins, or both, we measured the diameters of A6 and V6 using calipers in each patient, distinguishing between nondiseased segments and the presence of AO. Vascular dilatation was reported in relation to anatomy and referred to as “venous congestion” if affecting veins or “arterial dilatation” if affecting arteries. Additionally, we measured the left atrial size along its short and long axis and the right–left ventricular ratio using calipers.

#### 2.2.4. Degree of Pulmonary Artery Obstruction

Acute PE was defined as an opacification defect of pulmonary arteries, wholly or partially outlined by contrast medium. The radiologist recorded the extent of embolism as: unilateral or bilateral, its topography in terms of proximal, lobar, segmental, or subsegmental location, and the thrombotic burden according to the Qanadli CT obstruction index (CTOI) [20].

#### 2.2.5. Alveolar Opacity

AO was defined as the presence of ground-glass or consolidative opacities, or both. Each segment was also graded semi-quantitatively on a severity score for opacities ranging from 0 to 2 (0 = none, 1 = 1–50%, 2 = 51–100% of alveolar opacification), a previously established rating scheme that proved accurate and time-efficient [21]. Consequently, because each lung segment was assigned a semi-quantitative score ranging from 0 to 2, AO was measured according to 40 discrete severity levels (minimum severity score, 0; maximum score, 40 points/patient) [22].

### 2.3. Statistical Analysis

We reported the results as the number of subjects or segments (percentage) or mean (±SD), unless otherwise specified. Lung parenchyma was analyzed using qualitative and semi-quantitative variables per segment (vascular changes, categorical variable; the presence of alveolar opacities, ordinal variable (none, <50%, or >50%); the number of emboli and CTOI, quantitative; and A6 and V6 diameter in mm, quantitative). A bivariate statistical analysis was performed with two-tailed Chi-square, Wilcoxon two-sample, or Student’s *t*-test, as appropriate. Given the dataset’s moderate size, we refrained from performing multilevel analyses that could weaken the meaning of significant results. A *p*-value of >0.05 indicated a significant difference.

## 3. Results

### 3.1. Patient Characteristics

A total of 61 patients (23%) diagnosed with acute PE were included in the study (44 males and 17 females, mean age: 61.7 ± 14.2 years). The patients’ demographic characteristics are summarized in Table 1. The mean obstruction index (CTOI) was 13.3 ± 14.7%. The CTOI was low (<20%) in 43/61 (70.5%) patients, intermediate (20–37.5%) in 13/61 (21.3%), and high (≥40%) in 5/61 (8.2%). Four patients (6.6%) had fatal outcomes with a CTOI of 10, 20, 50, and 52.5%, respectively. Hypertension (27%), obesity (10%), and diabetes (8%) were the three most prevalent comorbidities. Only two patients (4%) had a pre-existing pulmonary condition of chronic obstructive pulmonary disease. Fifty-seven (93%) patients required hospital admission, of which 32 (65%) were transferred to the intensive care unit. Of the 32 intensive care unit patients, 24 (75%) required intubation. The 4/32 (12.5%) intensive care unit patients who died were all under mechanical ventilation.

With contrast-enhanced CT, all 61 patients showed signs of pneumonia in the form of AO (including ground-glass opacity and alveolar consolidation, combined) in at least one segment. The AO severity score per patient was 24 (18–32) (median (IQR)), indicating relatively severe disease in our cohort. Complete outcome data were available for 48 (78.7%) patients. In this group, the incidence of composite outcomes was 62.5%. Patients in the composite outcome subgroup had significantly more severe VC than survivors (mean number of segments with VC = 18 (14.3–19.8) vs. 13 (5–17)), *p* = 0.008, and AO than survivors (CT severity score = 30.5 (24.5–35.8) vs. 19 (6.8–21)), *p* < 0.001. Two (3.3%) patients had signs of pulmonary malignancy, and only one (1.6%) had pleural effusion.

### 3.2. Vascular Changes

We found vascular changes in all but 3/61 (4.9%) patients. Patients had a median (IQR) number of 16 (13–19) segments with vascular changes. In the quantitative vessel size analysis (Figure 2), A6′s (arterial) mean diameter (±SD) was 6.0 ± 1.4 mm in segments without AO and 6.5 ± 1.0 mm in segments with AO (*p* = 0.138). On the contrary, V6′s (venous) mean diameter was 6.2 ± 1.5 mm in segments without and 7.1 ± 1.0 mm in segments with AO (*p* = 0.031). Consequently, vascular remodeling primarily represented venous congestion (VC), as shown in Figure 3. Patients had VC in 14.7 ± 5.9 segments on average out of 20 pulmonary segments. Patients with fatal outcome had non-significantly more VC than survivors (CT severity score = 16.3 ± 3.1 vs. 14.6 ± 6), *p* = 0.502. Atrial size (mean ± standard deviation) was 37.9 ± 9.1 mm for the short axis and 69 ± 8.2 mm for the long axis. We found no correlation between VC severity and atrial size (R^2^ = 0.086). Thirty (49.2%) patients had an increased (>1.0) right–left ventricular ratio, without a correlation with VC (R^2^ = 0.015).

### 3.3. Anatomic Distribution of Pulmonary Emboli

There were 411 emboli in the 1220 segments analyzed (33.6%). The PE location is described in Table 2. Of the 329/411 (80%) segmental or subsegmental emboli, 58/329 (17.6%) were completely obstructive. Overall, segmental and subsegmental emboli were frequently located in the lower pulmonary regions (middle/lingula and lower lobes), i.e., 264/329 (80.2%) emboli, which is similar to what is known from non-COVID-19 cohorts [23], except in segments with VC and absent AO where emboli were located in the upper lobes for 80% (4/5) of cases. Conversely, in segments without VC and an absence of AO, only 24% (11/46) of emboli were located in the upper lobes.

### 3.4. Relationship between Pulmonary Embolism, Venous Congestion, and Alveolar Opacity

#### 3.4.1. Distribution of Pulmonary Embolism in Segments with Venous Congestion

Out of the 329 segmental and subsegmental emboli (excluding the main pulmonary artery and lobar emboli), 254 (77%) were located in segments presenting concomitant VC. Out of 58 completely obstructive emboli, 49 (84%) were located in segments having VC (Figure 4). The rate of segmental emboli tended to be higher in segments with VC without reaching statistical significance (28.3% vs. 23.1%, *p* = 0.07). More details are provided in Table 3, including the effect of AO severity. In segments with VC, the most common associated features were AO > 50% (449/896, 50%) and AO < 50% (413/896, 46%). Segments with VC and no AO were infrequently observed at 34/896 (3.8%).

#### 3.4.2. Distribution of Pulmonary Embolism Related to Alveolar Opacity

Of 329 segmental and subsegmental emboli, 278 (84.5%) were located in segments with AO. Figure 5 illustrates the distribution of PE among segments with and without VC, according to the presence or absence of AO. In the population overall, PE was significantly associated with AO (*p* = 0.047); VC likely drove this effect. When directly comparing emboli in segments with absent AO (21.8%) to those in segments with >50% AO (29.4%), the difference was even more significant (*p* = 0.03). AO and VC mutually reinforce the probability of PE. A clinical example is provided in Figure 6. Detailed data regarding the occurrence of PE in segments with VC, stratified by AO, and its severity (absent, <50%, or >50% alveolar opacification) are provided in Table 3.

In addition, Table 3 demonstrates that among the 75 emboli located in segments without VC, 46 were located within 200 (23%) segments without AO vs. 29 emboli within 124 (23.4%) segments with AO (*p* = 0.936), indicating that pneumonia-related hyperemia alone did not influence PE location significantly and is not the only driving mechanism. Furthermore, AO severity did not significantly influence the PE rate overall: 139 emboli were found within 513 (27.1%) segments with <50% AO vs. 139 emboli within 473 (29.4%) segments with >50% AO (*p* = 0.424). This finding suggests that VC drives the significant dependency of PE on AO since AO tended to affect the PE rate in segments with VC, although without reaching statistical significance (15% PE without AO vs. 28.9% with AO; *p* = 0.071), likely due to a lack of statistical power. The potential relationship between VC and PE in the same segment is supported by cases in which VC and PE coexist in a segment without AO (Figure 7).

We also noted that 15–23% of segments with no AO (i.e., no inflammation-mediated hyperemia) harbored segmental or subsegmental PE, 15% (5/34) of segments with concomitant VC, and 23% (46/200) without VC. Interestingly, four of the five (80%) emboli located in segments with no AO but present VC were associated with at least one pulmonary embolus in another segment with AO. Furthermore, 27 of the 46 (58.7%) emboli located in segments with neither AO nor VC were associated with at least one pulmonary embolus in another segment with AO, suggesting that hemodynamic phenomena from neighboring segments may be involved.

## 4. Discussion

Vascular dilatation has been suggested as a potential severity and prognostic marker early in the pandemic, including histopathologic studies [24]. We found vascular changes in affected veins rather than arteries and predicted the composite outcome of ICU admission or death in our work, including patients with proven SARS-CoV-2 infection and acute PE. In our cohort, a large majority (73.4%) of pulmonary segments had signs of VC, which aligns with previous studies [25,26]. Other publications, however, reported much lower rates of vascular dilatation, down to 18% [27,28]. In a systematic review, Lv et al. pooled the data of 1969 patients across 22 studies and reported a mean prevalence of the vascular enlargement sign of 69.4% [18]. This heterogeneity may be linked to the variable disease severity of included patients or methodological factors, such as the vascular dilatation’s definition itself. In this regard, most published works report vascular “congestion”, “enlargement”, or even “bronchovascular congestion” without specifying whether the affected compartment is mainly arterial, venous, or a combination of both [18,25,26,27,28]. Few reports provide details about arterial [29] or venous enlargement [6,30], and a recent study assumes that “most of the vessel diameters measured were of pulmonary arteries” [31]. On the other hand, the hypothesis that vascular remodeling mostly happens in the venous compartment has been suggested [6,14]; however, the evidence is lacking. We demonstrated that pulmonary veins dilate significantly in response to COVID-19 pneumonia, unlike the arterial diameter, raising questions related to underlying mechanisms because hypoxia typically causes vasoconstriction, such as in mosaic perfusion.

Just as VC, AO was highly prevalent in our study population and was reported in 80.8% of pulmonary segments, which aligns with our patients’ rather severe clinical status leading to intensive care unit admission in more than 50%. Indeed, Garg et al. reported a 66.9% overall prevalence of AO in a meta-analysis that included 6007 COVID-19 patients from 56 studies who underwent a chest CT within a week of symptom onset [32]. Acute PE distribution was significantly affected by AO overall. More specifically, the PE rate was affected by AO only in the presence of VC. Segments without VC had similar PE rates with or without AO, indicating that inflammation-related hyperemia is not the only mechanism governing embolus distribution. This theory is supported by the fact that AO’s severity (< 50% vs. > 50% involvement) did not affect PE prevalence either. This theory is consistent with Loffi et al., who found most emboli to be located in areas affected by pneumonia, but without a correlation between AO severity and PE [33].

Previous reports concluded that PE was more frequent in patients presenting VC [34]. Nonetheless, our study is the first to look into the detailed configuration of PE regarding AO and VC, providing new insights into possible underlying mechanisms. VC’s proposed etiologies are multiple: vasoactive due to pro-inflammatory cytokine release, inflammation-mediated hyperemia, capillary microthrombi formation, or dysregulation of physiological intrapulmonary arteriovenous anastomoses. In the case of purely vasoactive mechanisms, the net blood flow is not supposed to change as the venous compartment increases in volume—only the transit time is expected to be prolonged—contrary to observations based on DECT perfusion analysis [15,17]. In inflammation-mediated hyperemia, lung perfusion is likely to increase in segments presenting signs of COVID-19 pneumonia; however, oligemia was the most common finding (96%) in the study by Lang et al. [15]. On the other hand, hyperemia may explain PE’s association with VC under the influence of hemodynamic phenomena driving emboli to locations with high blood flow.

Furthermore, when capillary thrombi arise and block the downstream bed, arterial enlargement in response to the increased resistance is the anticipated sign. Following flow reduction, the drainage veins should be unchanged or even reduced in size. We observed exactly the opposite: no significant arterial enlargement and significant VC. Even a combination of multiple aforementioned mechanisms does not fully explain in vivo observations; however, the previously proposed hypothesis of arteriovenous anastomoses may help understand the process [6,7]. In the case of an arteriovenous shunt at the lobular level, blood flow deviates to the vein, causing oligemia in the downstream bed [15], which likely favors microthrombi formation in capillaries and increases blood flow due to lower resistance. According to the hypothesis of arteriovenous anastomosis dysregulation, creating a physiologic shunt, the vein receives arterial flow, and due to the venous capacitance, the vein appears enlarged. As this enlargement results from congestion rather than focal dilatation, we prefer the term “venous congestion” (VC). Intrapulmonary shunts, with or without the presence of concomitant PE, contribute to the patients’ hypoxia, whereas lung mechanics can remain relatively preserved [35]. Therefore, intrapulmonary shunts may be a potential research topic for patients with severe hypoxemia and preserved lung mechanics. Of course, different conditions, such as endothelial dysfunction, contribute to platelet aggregation and thrombus formation [36]. However, it is noteworthy that we did not find that the left atrial size depends on VC’s severity; such an effect can be challenging to determine in the absence of left heart failure.

Our study has several limitations. Firstly, this registry’s retrospective nature carries a risk of inclusion bias, and data regarding pretest probability (Wells score) or aspirin therapy prior to PE were unavailable. Furthermore, we only analyzed patients who underwent contrast-enhanced CT. Since most centers only perform contrast-enhanced CT in patients with dyspnea or respiratory distress and elevated D-dimer levels, our study population may include patients with acute respiratory distress syndrome and is not representative of COVID-19 pneumonia in the general population. Future studies are planned to address the prevalence and distribution of VC in patients with non-contrast CT. In addition, several of the effects we observed were close to reaching statistical significance, indicating potential interest for a larger sample size. However, according to the pre-specified study design [19], the sample size was well beyond the required number of emboli. Moreover, our analysis does not account for system DVT as an origin for PE, which may be unrelated to the variables under test in our cohort. Finally, we only included limited clinical, laboratory, and outcome analyses. Complete integration of such variables would fall beyond the scope of this work, and future studies are planned to report more details.

## 5. Conclusions

In conclusion, pulmonary vascular remodeling in COVID-19 pneumonia, often referred to as “dilatation” or “congestion”, primarily affects veins, and most pulmonary emboli were located in segments with venous congestion. Furthermore, alveolar opacity was not a predictor of PE in the absence of venous congestion. This finding supports the hypothesis that venous congestion may result from increased flow due to pulmonary arteriovenous anastomosis dysregulation.

## Figures and Tables

**Figure 1 biomedicines-10-01300-f001:**
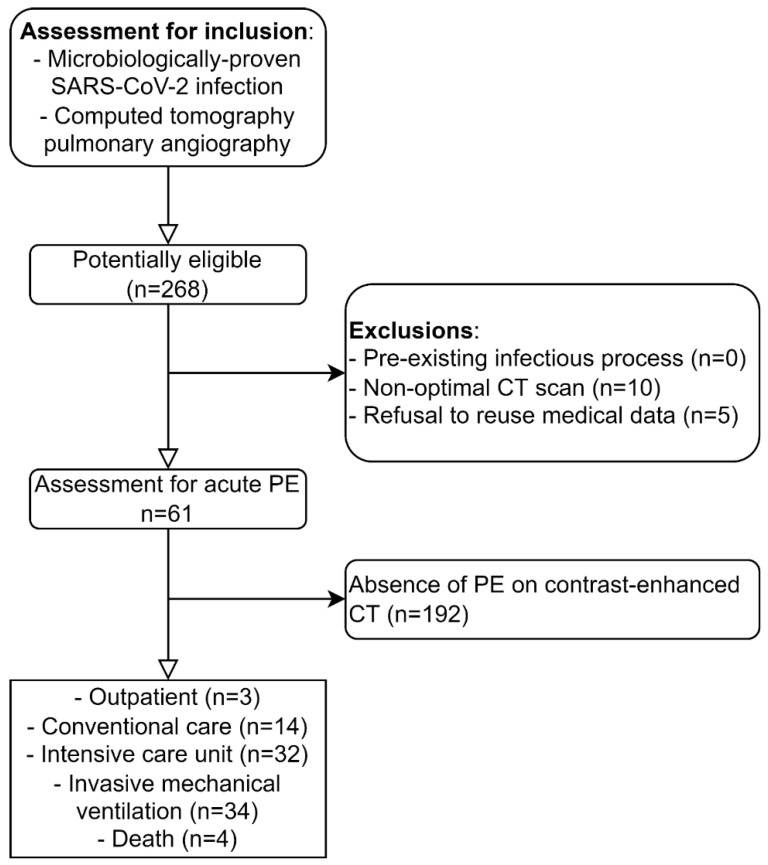
Study flow diagram.

**Figure 2 biomedicines-10-01300-f002:**
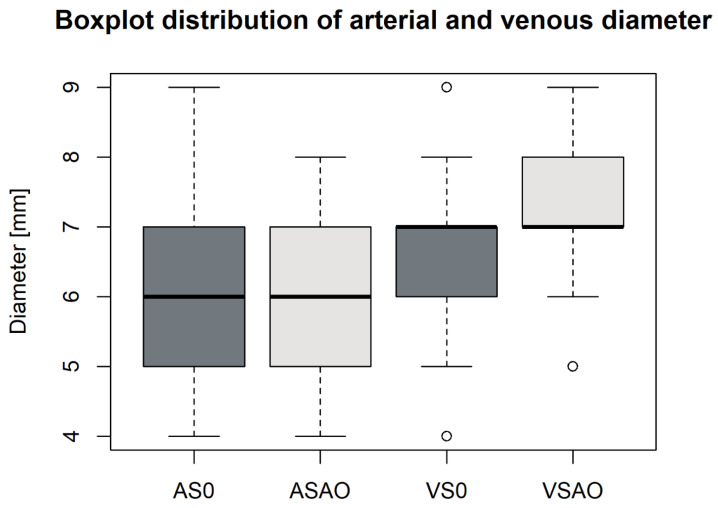
Boxplot distribution of arterial and venous diamters. Plots represent vascular diameters measured in A6 and V6 and classified according to the presence of COVID-19 pneumonia (AO) in the same lung segment. Venous diameters were significantly higher in diseased segments (*p* = 0.031). AO, alveolar opacity; AS0, arterial diameter in segments without alveolar opacity; ASAO, arterial diameter in segments with alveolar opacity; VS0, venous diameter in segments without alveolar opacity; and VSAO, venous diameter in segments with alveolar opacity.

**Figure 3 biomedicines-10-01300-f003:**
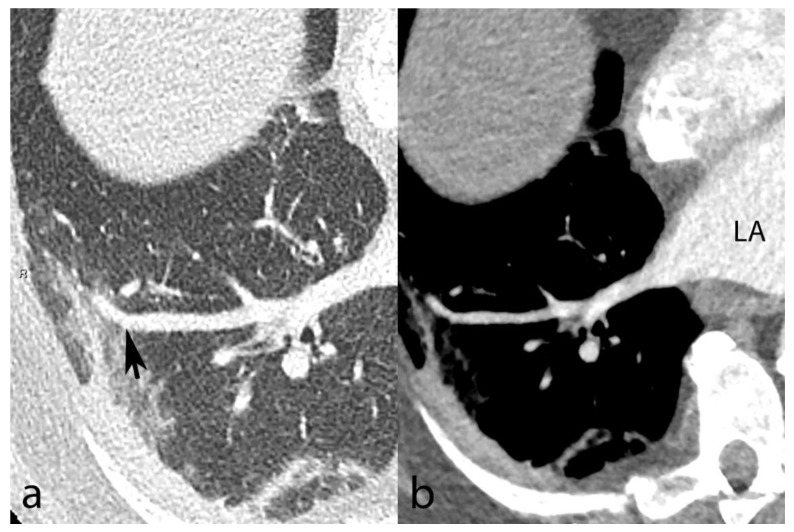
Contrast-enhanced chest CT from a 67-year-old male shows SARS-CoV-2 pneumonia with mixed peripheral ground-glass opacities and consolidation in the right lower lobe. Venous congestion is seen in the apical segment (panel (**a**), black arrow). A reformatted oblique axial image with mediastinal window settings (panel (**b**)) demonstrates the venous nature of the dilated vessel based on its connection to the left atrium (LA).

**Figure 4 biomedicines-10-01300-f004:**
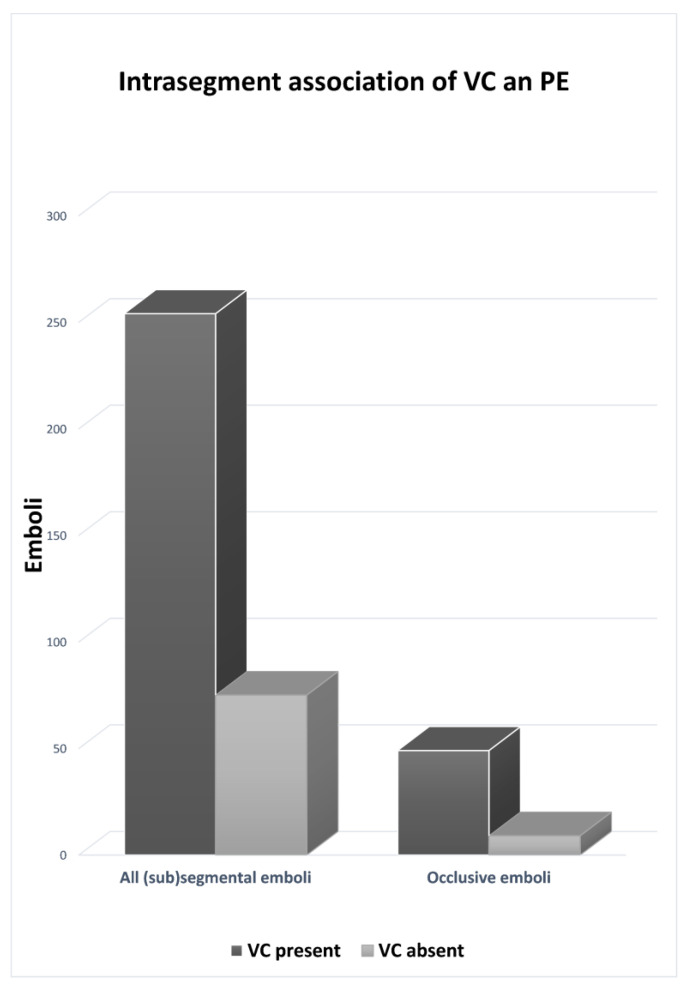
Intrasegment association of VC and PE. Out of 329 total segmental or subsegmental emboli, 77% (95% CI, 75–80%) were associated with VC in the same lung segment. Furthermore, out of 58 occlusive segmental or subsegmental emboli, 84% (95% CI, 80–89%) were associated with VC in the same lung segment.

**Figure 5 biomedicines-10-01300-f005:**
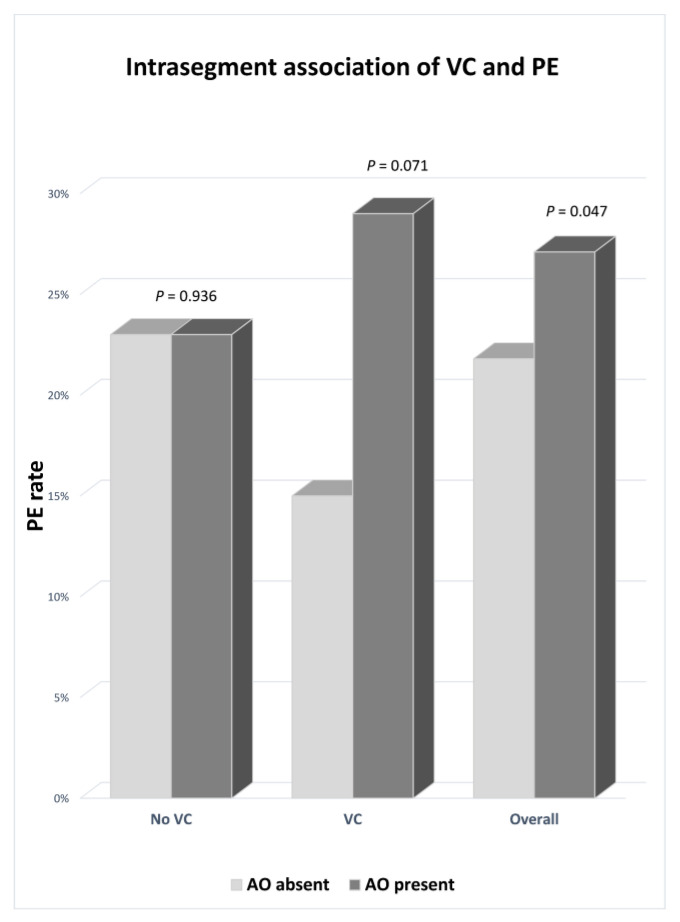
Intrasegment association of VC and PE. Graph shows the association between the rate of segmental or subsegmental PE found in segments with or without VC, respectively, stratified by the presence of AO. In the absence of VC, AO does not influence the PE rate. Conversely, when VC is present, AO markedly affects the probability of PE (n = 254 emboli). Statistical significance is reached only in the whole population due to higher statistical power (n = 329 emboli).

**Figure 6 biomedicines-10-01300-f006:**
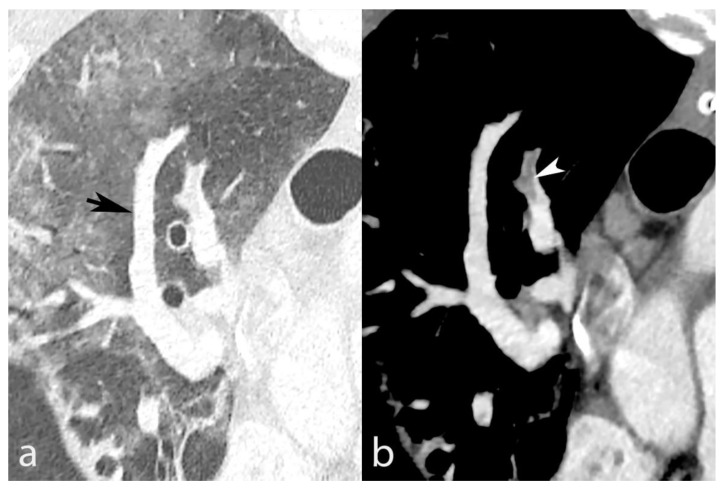
COVID-19 pneumonia with AO and VC (panel (**a**), black arrow) in a 63-year-old man. Acute PE (panel (**b**), white arrowhead) was found in the right upper lobe. The concomitant presence of PE and VC raises the question of arteriovenous anastomosis dysregulation increasing blood flow in the vein despite the arterial obstruction.

**Figure 7 biomedicines-10-01300-f007:**
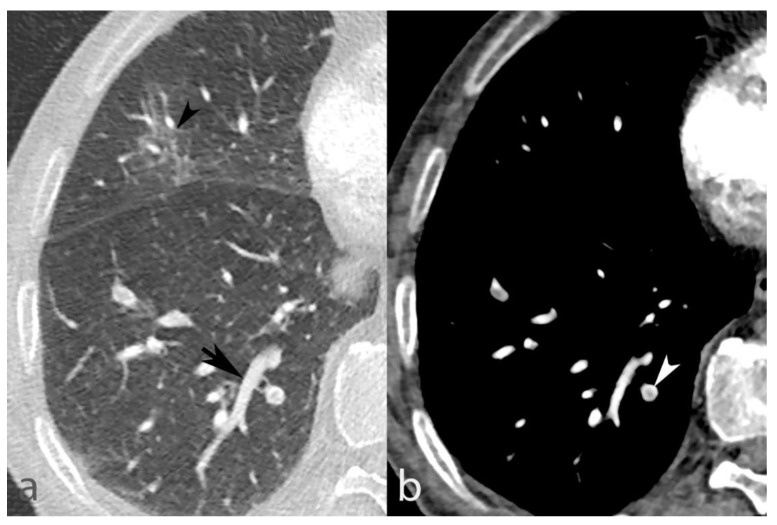
Mild AO in the right middle lobe (panel (**a**), black arrowhead), without associated VC in an 85-year-old man. On the other hand, distinct VC is seen in the right lower lobe (panel **a**, black arrow) with concomitant PE (panel (**b**), white arrowhead), supporting the potential role of VC in PE’s distribution.

**Table 1 biomedicines-10-01300-t001:** Patient characteristics.

Characteristic	Patients (n = 61)
**Mean age ± SD (y)**	61.7 ± 14.2
**Sex, n (%)**	
Male	44 (72)
Female	17 (28)
**Comorbidities, n (%) ^1^**	
Heart failure	2 (4)
Hypertension	13 (27)
Obesity	5 (10)
Diabetes	4 (8)
History of PE	2 (4)
History of malignancy	3 (6)
**Laboratory mean, ± SD**	
D-Dimer (ng/mL) ^2^	8516 ± 11175
Troponin (μg/L) ^3^	46 ± 94
PaO2 (mmHg) ^4^	70 ± 29
CRP (mg/L) ^5^	150 ± 113
**Type of care, n (%) ^1^**	
Ambulatory	3 (6)
Conventional care	14(29)
Critical care (ICU)	32 (65)
Invasive mechanical ventilation	24 (49)
**Anticoagulationthromboprophylaxis, n (%) ^6^**	14 (32)
**Death, n (%) ^1^**	4 (8)

^1.^ Data available for 49 patients. ^2.^ Data available for 37 patients. ^3.^ Data available for 32 patients. ^4.^ Data available for 27 patients. ^5.^ Data available for 35 patients. ^6.^ Data available for 44 patients.

**Table 2 biomedicines-10-01300-t002:** Anatomical distribution of pulmonary emboli (n = 329) in upper, middle, and lower pulmonary segments (20 segments in 61 patients).

Anatomical Level	Number of Emboli
Troncular	23 (6%)
Lobar	59 (14%)
Segmental/sub-segmental	329 (80%)
** Upper lobes*	*65 (16%)*
*# Middle lobe/Lingula*	*34 (8%)*
*° Lower lobes*	*230 (56%)*

* Upper lobes included three segments: apical, posterior, and anterior on the right side; apico-posterior (counted as two segments) and anterior on the left side. # The middle lobe/lingula included two segments: medial middle lobe and lateral middle lobe on the right side; superior lingular and inferior lingular on the left side. ° Lower lobes included five segments: superior, medial, anterior, lateral, and posterior on the right side; superior, anteromedial (counted as two segments), lateral, and posterior on the left side.

**Table 3 biomedicines-10-01300-t003:** Rate of segmental and subsegmental PE in segments with and without concomitant VC, stratified by AO severity (n = 329 segmental emboli in 1200 pulmonary segments).

Emboli/Segments	VC +	VC −	Total
No AO	5/34 (15%)	46/200 (23%)	51/234 (21.8%)
AO < 50%	115/413 (28%)	24/100 (24%)	139/513 (27.1%)
AO > 50%	134/449 (30%)	5/24 (21%)	139/473 (29.4%)
**Total**	254/896 (28.3%)	75/324 (23.1%)	329/1220

Data are presented as emboli/segments (%). The highest incidence of PE (i.e., 30%) was found in segments with VC and concomitant AO > 50%, while the lowest (i.e., 21%) was found in segments without VC but concomitant AO > 50%. PE, pulmonary embolism; VC +, venous congestion present; VC −, venous congestion absent; and AO, alveolar opacity.

## Data Availability

The data presented in this study are available upon request from the corresponding author. The data are not publicly available because they contain information that could compromise research participant privacy.

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
