# Peer review of "Acute Pulmonary Embolism in COVID-19: A Potential Connection between Venous Congestion and Thrombus Distribution"

_biomedicines, 2022, doi:10.3390/biomedicines10061300_

Round 1

Reviewer 1 Report

The authors studied the relations of AO, CV & PE distribution in patients with COVID-19.

They conclude that there is an association between VC and PE, that the mechanism is mainly venous and not arterial, and that there is no association between the severity of AO and PE. The paper is well written, yet can be improved:

  1. The authors divided the patients to no AO, <50% AO and >50% AO. This is a too broad division. What is the then the true difference between 40% and 60% AO? I would suggest that the authors divide the cohort into quartiles or tertiles. This may result in better definition of the relations of AO & PE.
  2. Can the authors provide data on left atrial size, volume or pressure in these patients? is it possible that VC is related to increased LA pressure? 
  3. The authors did not provide any data on prior aspirin treatment of anticoagulation either pre study or post study...
  4. Can you elaborate on the clinical significance of the study? what are the implications for the clinician? 
  5. Add a "conclusion" paragraph at the end of the discussion. 

Author Response

Reviewer 1

Comments and Suggestions for Authors

Thank you for the time spent reviewing our manuscript and for the helpful suggestions that helped us improve the manuscript. Please find our responses below.

The authors studied the relations of AO, CV & PE distribution in patients with COVID-19.

They conclude that there is an association between VC and PE, that the mechanism is mainly venous and not arterial, and that there is no association between the severity of AO and PE. The paper is well written, yet can be improved:

  1. The authors divided the patients to no AO, <50% AO and >50% AO. This is a too broad division. What is the then the true difference between 40% and 60% AO? I would suggest that the authors divide the cohort into quartiles or tertiles. This may result in better definition of the relations of AO & PE.

Response: We thank the reviewer for commenting on the semi-quantitative AO scheme because the way we described it in the methods might be confusing. The AO rating scheme we used was developed specifically as a tool to assess COVID-19 pneumonia severity, published as the pandemic started spreading (https://doi.org/10.1148/ryct.2020200047). This scheme assigns each pulmonary segment to one of three categories: no AO, <50%, or >50% of the segment affected by AO. Per patient, this translates into 40 semi-quantitative levels. We have revised the Methods section for better clarity.

  1. Can the authors provide data on left atrial size, volume or pressure in these patients? is it possible that VC is related to increased LA pressure? 

Response: this is an insightful comment from the reviewer, for which we are grateful. We happen to have left atrial size date and tried to find out whether or not there is a correlation with VC. However, the R2 coefficient showed a poor fit. This is an interesting finding that makes sense in the way that left cardiac function is not necessarily failing in COVID-19 pneumonia with PE. We have added this finding in the results and updated the discussion.

  1. The authors did not provide any data on prior aspirin treatment of anticoagulation either pre study or post study...

Response: thank you for commenting on this critical issue. While table 1 indicated that 14 patients were under thromboprophylaxis, the term was now updated to "Anticoagulation thromboprophylaxis" to avoid any misunderstanding. Unfortunately, information regarding aspirin was not available, which we mentioned in the limitations.

  1. Can you elaborate on the clinical significance of the study? what are the implications for the clinician?

Response: the reviewer is correct; direct implications for the clinician may not be straightforward since this is a descriptive study trying to discuss possible physiopathological mechanisms. The clinically relevant finding we are pointing out is the possibility of intrapulmonary arteriovenous anastomoses playing a role in thrombus distribution, which, as a result, implies a shunt that contributes to hypoxia – with or without PE. Physiological and pathological states can trigger such shunts. However, discussing treatment targets directed at arteriovenous anastomoses would be premature and fall beyond our study's aims. We expanded the discussion to outline this concept.

  1. Add a "conclusion" paragraph at the end of the discussion.

Response: We thank the reviewer for pointing out this issue. We added a conclusion at the end of the discussion.

Reviewer 2 Report

I read this manuscript with great interest. It is a well-designed and well-written study.

I have some comments:

  • Please add info about respiratory diseases in methods and tables.
  • Please add Wells score in methods and tables.
  • Please specify whether sample size was evaluated in statistical analysis section.
  • Please correct several minor English errors throughout the manuscript.

Author Response

Reviewer 2

Comments and Suggestions for Authors

Thank you for the time spent reviewing our manuscript and for the helpful suggestions that helped us improve the manuscript. Please find our responses below.

I read this manuscript with great interest. It is a well-designed and well-written study.

I have some comments:

  1. Please add info about respiratory diseases in methods and tables.

Response: thank you for suggesting expanding on respiratory diseases. We updated the methods and results sections accordingly.

  1. Please add Wells score in methods and tables.

Response: We are grateful for this constructive suggestion. Unfortunately, we could not reliably calculate the Wells score as this was not routinely done before the CTs were ordered.

  1. Please specify whether sample size was evaluated in statistical analysis section.

Response: We thank the reviewer for drawing our attention to an essential component of clinical research. The sample size was indeed used and reported in ref 19. We have summarized the results of sample size calculation in the methods for better transparency.

  1. Please correct several minor English errors throughout the manuscript.

Response: thank you for kindly letting us know. We have carefully edited typos throughout the manuscript.

Reviewer 3 Report

Venous congestion and pulmonary embolism are different entities from a pathophysiological and clinical point of view. This delimitation is not clear from the article. Pulmonary thromboembolism ”affects” the pulmonary arterial territory. The terms ”Venous” and ”vascular” congestion are not very well defined in this manuscript. The title itself is a source of confusion.

Pulmonary thromboembolism affects the pulmonary arterial territory.

The article only expresses the radiologist's point of view without correlating with the clinic.

The number of studied patients is too small.

Some of my observations are marked into the text

Author Response

Reviewer 3

Comments and Suggestions for Authors

Thank you for the time spent reviewing our manuscript and for the helpful suggestions that helped us improve the manuscript. Please find our responses below.

Venous congestion and pulmonary embolism are different entities from a pathophysiological and clinical point of view. This delimitation is not clear from the article. Pulmonary thromboembolism" affects" the pulmonary arterial territory. The terms" Venous" and" vascular" congestion are not very well defined in this manuscript. The title itself is a source of confusion.

  1. Pulmonary thromboembolism affects the pulmonary arterial territory.

Response: we entirely agree with the reviewer that pulmonary thromboembolism affects the arterial territory. Additionally, we believe that the arterial and venous compartments are connected, and the flow/pressure dynamics of one can influence the other; demonstrating this type of effect in COVID-19 is this work's aim. This is known from acute PE studies; massive PE can cause left atrial volume to drop, and even left ventricular preload to decrease (doi: 10.2214/AJR.07.3715). This means that disorders in the pulmonary arterial compartment can even influence the systemic arterial compartment. In this context, we thank the reviewer for pointing out a possible confusion in the title: we have now revised it.

Furthermore, we have clarified the question of "vascular" or "venous" congestion. In our study, the fact that veins are more affected by dilatation than arteries is a study result. For this reason, we introduce the specific term "venous congestion" only in the results section and refer to "vascular dilatation" (as we do not a priori know whether it is related to arteries, veins, or both) before. We have edited the terminology throughout the manuscript.

  1. The article only expresses the radiologist's point of view without correlating with the clinic.

Response: the reviewer is correct; direct implications for the clinician may not be straightforward since this is a descriptive study trying to discuss possible physiopathological mechanisms. The clinically relevant finding we are pointing out is the possibility of intrapulmonary arteriovenous anastomoses playing a role in thrombus distribution, which, as a result, implies a shunt that contributes to hypoxia – with or without PE. Such shunts can be triggered by physiological and pathological states. However, discussing treatment targets directed at arteriovenous anastomoses would be premature and fall beyond our study's aims. We expanded the discussion to outline this concept.

  1. The number of studied patients is too small.

Response: we agree with the reviewer and acknowledge that a larger sample size might be interesting, as pointed out in the limitations. On the other hand, our sample size calculation published in the study design indicated that we had to analyze at least 182 lung segments with PE to draw conclusions regarding the influence of lung inflammation on thrombus distribution. We ended up with 329 segmental/subsegmental emboli (and 59 lobar, 23 troncular emboli), exceeding the required sample size. This was explained better in the limitations paragraph.

  1. Some of my observations are marked into the text

Response: We thank the reviewer for specific suggestions regarding certain concepts' phrasing and pointing out spelling issues. We have carefully checked and edited the manuscript.

Reviewer 4 Report

Thank you for the opportunity to review this manuscript. This paper analyzed an interesting paradigm i.e. correlation of venous congestion and alveolar opacities with PE. The presence of downstream venous congestion causing PE upstream in arterial circulation without significant arterial dilation is rather enigmatic and not well explained by existing hypothesis. Authors’ hypothesis here trying to explain this phenomenon has merits.

The data analysis however needs to be addressed in this manuscript. The number of patients in this study is small and multiple comparisons without correction can lead to type I error. I would recommend that authors correct the analysis for multiple comparisons. It does not need be as conservative as Bonferroni correction as this is a preliminary study but other less conservative methods can be employed.

It would also be interesting to know if clinical severity of pulmonary disease have any effects on the outcome. It may be difficult given the small number of patients in this dataset but it maybe still be worthwhile to analyze this factor.

Also, this study does not take into account the PE secondary to systemic DVTs (commonly seen with COVID-19) which may be unrelated to the factors in analysis here. This needs to mentioned in the limitation section.

Author Response

Reviewer 4

Comments and Suggestions for Authors

Thank you for the time spent reviewing our manuscript and for the helpful suggestions that helped us improve the manuscript. Please find our responses below.

  1. Thank you for the opportunity to review this manuscript. This paper analyzed an interesting paradigm i.e. correlation of venous congestion and alveolar opacities with PE. The presence of downstream venous congestion causing PE upstream in arterial circulation without significant arterial dilation is rather enigmatic and not well explained by existing hypothesis. Authors' hypothesis here trying to explain this phenomenon has merits.

Response: We thank the reviewer for this kind comment.

  1. The data analysis however needs to be addressed in this manuscript. The number of patients in this study is small and multiple comparisons without correction can lead to type I error. I would recommend that authors correct the analysis for multiple comparisons. It does not need be as conservative as Bonferroni correction as this is a preliminary study but other less conservative methods can be employed.

Response: the reviewer correctly underlines that datasets of moderate size must be handled with care when it comes to multiple comparisons. When performed, multiple comparisons require the application of correction methods. To avoid uncertainties related to multilevel testing, we refrained from performing them. This has explained in the methods section.

  1. It would also be interesting to know if clinical severity of pulmonary disease have any effects on the outcome. It may be difficult given the small number of patients in this dataset but it maybe still be worthwhile to analyze this factor.

Response: We thank the reviewer for this helpful suggestion. Indeed, both AO and VC significantly predicted outcomes when looking at the composite outcome of ICU admission or death. This was added to the results.

  1. Also, this study does not take into account the PE secondary to systemic DVTs (commonly seen with COVID-19) which may be unrelated to the factors in analysis here. This needs to mentioned in the limitation section.

Response: We are grateful for this insightful comment. This factor is indeed important and cannot be assessed accurately in our study. We have revised the limitations section accordingly.

Round 2

Reviewer 1 Report

I have no further comments. The authors hac=ve answered all my comments appropriately.

Author Response

Thank you for your time.

Reviewer 3 Report

 I have reviewed the new version of the manuscript, and although many improvements have been made, many things remain unclear. I marked in blue with some of these phrases. Although the article makes strict reference to the imaging changes, clinical and pathophysiological data are also mentioned but those data are not clearly integrated in the computer tomographic examination.

Ussually the consequences of pulmonary embolism are reflected on right cavities  of the heart. Why only left atrium diameters were measured? Nothing about right cavities. 

Author Response

Reviewer 3

Comments and Suggestions for Authors

Thank you for the time spent reviewing our revised manuscript. We have further impoved the article based on your advice. Please find our responses below.

 I have reviewed the new version of the manuscript, and although many improvements have been made, many things remain unclear. I marked in blue with some of these phrases. Although the article makes strict reference to the imaging changes, clinical and pathophysiological data are also mentioned but those data are not clearly integrated in the computer tomographic examination.

We have revised the potentially unclear sentences marked in blue.

Ussually the consequences of pulmonary embolism are reflected on right cavities  of the heart. Why only left atrium diameters were measured? Nothing about right cavities. 

The reviewer is correct, the consequences of severe pulmonary emblism can impact right ventricular function. We have now added right-to-left ventricular data and the correlation coefficient with venous congestion.

If there are any other issues, please let us know. We will be happy to answer your questions.

Reviewer 4 Report

Thank you for addressing the comments appropriately.

Author Response

Thank you for your time.

Round 3

Reviewer 3 Report

I have read the new version of the manuscript.

I have marked some of my observations into the text.

Author Response

We thank the reviewer for spending additional time commenting on our manuscript. We have expanded on the issue of hypercoagulability and acknowledge that mechanisms are complex and incompletely understood. We have clarified the status of references 19 and 3.